# Commensal bacteria augment *Staphylococcus aureus* infection by inactivation of phagocyte-derived reactive oxygen species

Josie F. Gibson [1,2,3]☯, Grace R. Pidwill [1,2]☯, Oliver T. Carnell [1,2], Bas G. J. Surewaard [4,5], Daria Shamarina [1,2], Joshua A. F. Sutton [1,2], Charlotte Jeffery [1], Aurélie Derré-Bobillot [6], Cristel Archambaud [6], Matthew K. Siggins [7,8], Eric J. G. Pollitt [1,2], Simon A. Johnston [2,3,9], Pascale Serror [6], Shiranee Sriskandan [7,8], Stephen A. Renshaw [2,3,9]*, Simon J. Foster [1,2]*

1 Department of Molecular Biology and Biotechnology, University of Sheffield, Sheffield, United Kingdom, 2 Florey Institute, University of Sheffield, Sheffield, United Kingdom, Sheffield, United Kingdom, 3 The Bateson Centre, University of Sheffield, Sheffield, United Kingdom, 4 Snyder Institute for Chronic Diseases, Cumming School of Medicine, University of Calgary, Calgary, Alberta, Canada, 5 Department of Microbiology, Immunology and Infectious Diseases, Cumming School of Medicine, University of Calgary, Calgary, Alberta, Canada, 6 Université Paris-Saclay, INRAE, AgroParisTech, Micalis Institute, Jouy-en-Josas, France, 7 Department of Infectious Disease, Imperial College London, London, United Kingdom, 8 MRC Centre for Molecular Bacteriology and Infection, Imperial College London, London, United Kingdom, 9 Department of Infection, Immunity and Cardiovascular disease, Medical School, University of Sheffield, Sheffield, United Kingdom

☯ These authors contributed equally to this work.
* s.a.renshaw@sheffield.ac.uk (SAR); s.foster@sheffield.ac.uk (SJF)

**Data Availability Statement:** All the data are deposited on the ORDA database, link:10.15131/shef.data.15134709

## Abstract

*Staphylococcus aureus* is a human commensal organism and opportunist pathogen, causing potentially fatal disease. The presence of non-pathogenic microflora or their components, at the point of infection, dramatically increases *S. aureus* pathogenicity, a process termed augmentation. Augmentation is associated with macrophage interaction but by a hitherto unknown mechanism. Here, we demonstrate a breadth of cross-kingdom microorganisms can augment *S. aureus* disease and that pathogenesis of *Enterococcus faecalis* can also be augmented. Co-administration of augmenting material also forms an efficacious vaccine model for *S. aureus*. *In vitro*, augmenting material protects *S. aureus* directly from reactive oxygen species (ROS), which correlates with *in vivo* studies where augmentation restores full virulence to the ROS-susceptible, attenuated mutant *katA ahpC*. At the cellular level, augmentation increases bacterial survival within macrophages via amelioration of ROS, leading to proliferation and escape. We have defined the molecular basis for augmentation that represents an important aspect of the initiation of infection.

## Author summary

*S. aureus* is a commensal inhabitant of the human skin and nares. However, it can cause serious diseases if it is able to breach our protective barriers such as the skin, often via wounds or surgery. If infection occurs via a wound, this initial inoculum contains both

**Funding:** SAR and SJF were supported by Medical Research Council (MRC) grant MR/R001111/1 (www.mrc.ukri.org), SAJ was supported by MRC grant MR/J009156/1 (www.mrc.ukri.org), and SS by MRC grant MR/L008610/1 (www.mrc.ukri.org). PS were supported by the French National Research Institute for Agriculture, Food, and Environment (INRAE) funding (www.inrae.fr). Imaging was performed in the Sheffield RNAi Screening Facility (Wellcome Trust grant reference number 084757 (www.wellcome.org)) and the Wolfson Light Microscopy Facility (MRC grant G0700091 (www.mrc.ukri.org) and Wellcome grant 077544/Z/05/Z (www.wellcome.org)). The funders had no role in study design, data collection and analysis, decision to publish, or preparation of the manuscript.

**Competing interests:** The authors have declared that no competing interests exist.

**Abbreviations:** CFU, colony forming units; CGD, chronic granulomatosis disease; dpf, days post-fertilisation; GFP, green fluorescent protein; HK, heat-killed; hpi, hours post-infection; HOCl, hypochlorous acid; $H_2O_2$, hydrogen peroxide; iNOS, inducible nitric oxide synthase; MDMs, monocyte-derived macrophages; MPO, myeloperoxidase; NOX2, NADPH oxidase; PGN, peptidoglycan; RNS, reactive nitrogen species; ROS, reactive oxygen species; Tg, transgenic.

the pathogen, other members of the microflora and also wider environmental microbes. We have previously described "augmentation", whereby this other non-pathogenic material can enhance the ability of *S. aureus* to lead to a serious disease outcome. Here we have determined the breadth of augmenting material and elucidated the cellular and molecular basis for its activity. Augmentation occurs via shielding of *S. aureus* from the direct bactericidal effects of reactive oxygen species produced by macrophages. This initial protection enables the effective establishment of *S. aureus* infection. Understanding augmentation not only explains an important facet of the interaction of *S. aureus* with our innate immune system, but also provides a platform for the development of novel prophylaxis approaches.

## Introduction

*Staphylococcus aureus* exists in a polymicrobial environment, primarily as a human commensal organism [1–3], but can also cause disease after a breach in host defences, often via localised tissue injury [4]. *S. aureus* causes a spectrum of disease, from minor skin infections to life-threatening bacteraemia: infections that are increasingly difficult to treat due to antibiotic resistance [5]. Human innate immune defences, primarily phagocytes, play a crucial role in preventing serious *S. aureus* disease. However, during infection *S. aureus* can reside within, and escape from, an intraphagocyte niche [6–8]. Similar to other intracellular pathogens [9,10], this can lead to a population bottleneck, where most bacteria are effectively killed by phagocytes, but a small proportion survive, enabling continued infection [11]. This results in the emergence of clonal bacterial populations, which expand from the small numbers surviving the population bottleneck. In the murine sepsis model, liver-resident macrophages known as Kupffer cells are the basis of this population bottleneck and subsequent bacterial clonality [12,13]. Macrophages are crucial for defence against *S. aureus*, exposing bacteria to an array of bactericidal mechanisms, including ROS, deleterious enzymes and antimicrobial peptides [14]. After phagocytosis, NADPH oxidase (NOX2) produces superoxide ($O_2^-$) [15], which is converted to hydrogen peroxide ($H_2O_2$) and hydroxyl radical ($\cdot$OH). Hypochlorous acid (HOCl) is generated from $H_2O_2$ via the enzyme myeloperoxidase (MPO) [16]. Reactive nitrogen species (RNS) are produced by inducible nitric oxide synthase (iNOS), creating nitric oxide (NO$\cdot$) which can then react with $O_2^-$ to form peroxynitrite (ONOO$^-$) [17]. All reactive species cause bacterial damage, but HOCl and $H_2O_2$ may be key *in vivo*, as both are efficacious against biofilms [18,19]. *S. aureus* uses several approaches to resist ROS/RNS: two superoxide dismutases detoxify $O_2^-$ [20], catalase removes $H_2O_2$ [21], alkyl hydroperoxidase acts to reduce $H_2O_2$, ONOO$^-$ and organic peroxides [22], and staphylococcal peroxidase inhibitor (SPIN) inhibits MPO therefore blocking HOCl formation [23]. Many *S. aureus*-ROS studies focus on neutrophils, since chronic granulomatosis disease (CGD) highlights ROS as vital in neutrophil bacterial clearance [24]. Nevertheless, ROS are also important in tissue macrophages [25] for defence against *S. aureus* [26].

Augmentation is a recently described phenomenon whereby human skin commensals enhance *S. aureus* pathogenesis [12]. *S. aureus* bloodstream infection in mice can be augmented by either Gram-positive commensals, their purified peptidoglycan (PGN) or a natural mix of skin flora [12]. In this example of microbial crowdsourcing, only *S. aureus* benefits, not the non-pathogenic commensals, which succumb. During murine sepsis, augmenting material is co-phagocytosed with *S. aureus* in Kupffer cells, resulting in increased bacterial survival and the subsequent formation of clonal liver microabscesses [12], with the potential to seed other

organs in the body [13]. In Kupffer cells, augmentation is associated with reduced oxidation and, importantly, augmentation is not observed in transgenic mice lacking functional NOX2, defining a pivotal role for ROS in this phenomenon. However, major signalling receptor-mediated mechanisms (including NOD1, NOD2, TLRs and the inflammasome) did not account for augmentation [12]. To elucidate the molecular mechanism(s) underpinning augmentation, we sought to define the breadth of materials able to enhance *S. aureus* infection and investigate whether augmentation occurs for other human pathogens. Using *in vitro* and *in vivo* studies, we demonstrate that the molecular basis for augmentation is absorption of ROS by augmenting material, shielding *S. aureus* from macrophage-mediated killing.

## Results

### A broad range of pathogen-derived materials augment *S. aureus* infection

Previously, we have shown that *S. aureus* pathogenesis can be augmented by live Gram-positive skin commensals, purified PGN, or natural skin flora [12]. To determine the breadth of material able to augment *S. aureus* pathogenesis, we used the murine sepsis model and co-injection of a low *S. aureus* infectious dose with potential augmenting materials. Increased bacterial numbers in the liver is a key marker of augmented infection, with accompanying weight loss and/or increased kidney bacterial load in severe cases [12]. We first tested Gram-negative bacteria *Escherichia coli* and *Roseomonas mucosa*, as part of the human microflora [27,28], with heat-killed (HK) *Micrococcus luteus* as a positive control. Addition of HK *M. luteus*, *E. coli* or *R. mucosa* significantly increased *S. aureus* counts in the liver in comparison to *S. aureus*-only infected mice (Figs 1A–1C, 1E and S1A–S1C). On average *S. aureus* liver counts are greatly increased from the inocula (of $1x10^6$ CFU) to $1.25x10^8$ CFU in augmented infections, in comparison to $1x10^6$ in control infections. Interestingly, *E. coli* benefits from the presence of *S. aureus*, with an increase in *E. coli* counts in the liver, although these CFU counts are reduced in comparison to the injected *E. coli* inoculum (Fig 1B). In order to assess whether cross-kingdom materials could augment staphylococcal infection, we tested HK *Cryptococcus neoformans* and live fungi in the murine sepsis model. Addition of HK *C. neoformans* significantly increased *S. aureus* liver numbers (Figs 1D, 1E and S1D). In contrast, *Saccharomyces cerevisiae*, an occasional human commensal which rarely becomes pathogenic [29,30], did not increase *S. aureus* liver or kidney numbers, but did enhance mouse weight loss (Figs 1E and S1E–S1G). Together these data demonstrate that *S. aureus* pathogenicity can be enhanced by a wider range of microorganisms than has previously been shown.

### Can infection with other human pathogens be augmented?

Next, we tested whether augmenting material was able to increase the virulence of a range of human pathogens: *Enterococcus faecalis*, an opportunist pathogen capable of residing within macrophages [31]; *Streptococcus pneumoniae*, which is able to survive within phagocytes and experiences a population bottleneck which seeds further infection [32]; *Pseudomonas aeruginosa*, which survives within macrophages [33]; and *Streptococcus pyogenes*, which can survive and escape from within host cells [34]. During murine sepsis, live *M. luteus* augmented *E. faecalis* infection (of $5x10^7$ CFU) with a significant increase in liver and lung bacteria compared to *E. faecalis* alone (Figs 1F, 1G and S1H–S1L). Furthermore, *M. luteus* PGN augmented a larger *E. faecalis* inoculum ($1x10^8$ CFU), although it did not at $5x10^7$ CFU (S1M–S1R Fig). In both cases of *E. faecalis* augmentation, the liver bacterial number never increased higher than the inoculum. Pathogenesis of *S. pneumoniae* and *P. aeruginosa* was not increased by the presence of *M. luteus* PGN in sepsis models (Figs 1G and S2A–S2F). Also *M. luteus* PGN did not alter mouse weight or *S. pyogenes* numbers in an intra-muscular leg infection model (S2G and

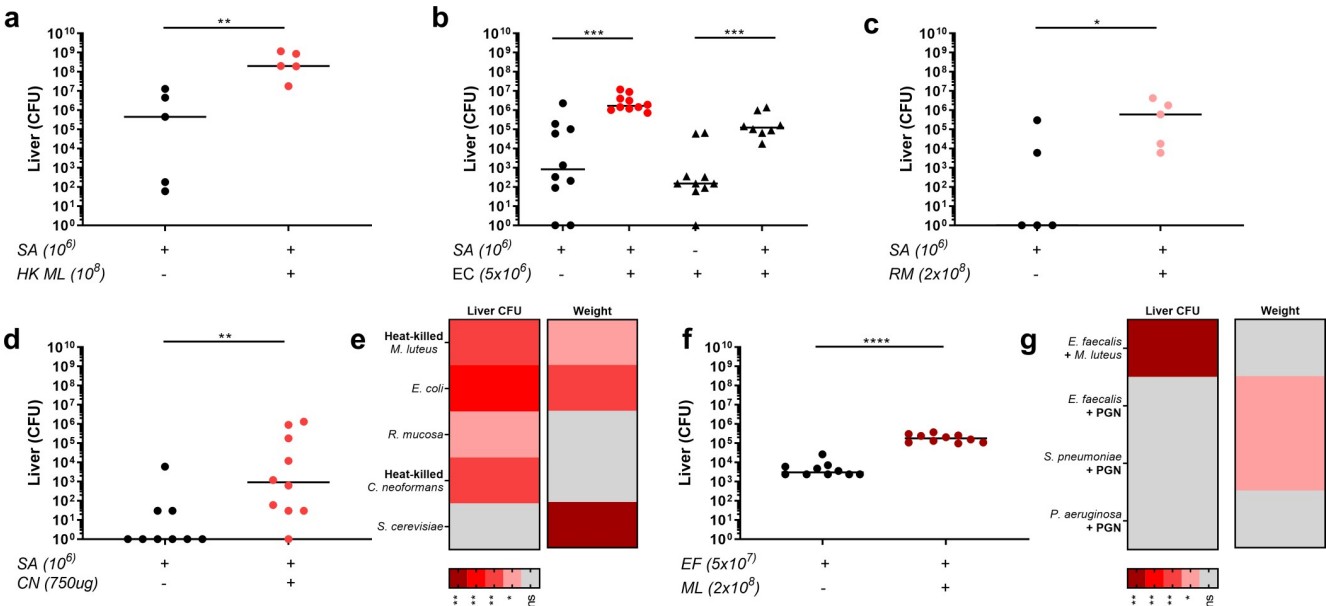

**Fig 1. Breadth of the augmentation phenomenon, from *S. aureus* to other pathogens.** Co-injection of low dose *S. aureus* (SA 1x10⁶ CFU) with heat-killed *M. luteus* (HK ML equivalent of 1x10⁸ CFU) into mice: liver CFU (n = 5 per group, median value shown, two-tailed Mann-Whitney test), **p<0.008 **B** Co-injection of low dose *S. aureus* (SA 1x10⁶ CFU) with *E. coli* (EC 5x10⁶ CFU) into mice: liver CFU, *S. aureus*, circles; *E. coli* triangles (n = 8–10 per group, median value shown, two individual two-tailed Mann-Whitney tests), ***p<0.0003 **C** Co-injection of low dose *S. aureus* (SA 1x10⁶ CFU) with *R. mucosa* (RM 2x10⁸ CFU) into mice: liver CFU (n = 5 per group, median value shown, two-tailed Mann-Whitney test), *p<0.05 **D** Co-injection of low dose *S. aureus* (SA 1x10⁶ CFU) with heat-killed *C. neoformans* (CN 750 µg) into mice: liver CFU (n = 9–10 per group, median value shown, two-tailed Mann-Whitney test), **p<0.006 **E** Summary heat-map for microorganisms tested for augmenting ability in *S. aureus* infection, showing significant changes in liver CFUs and weight change **F** Co-injection of low dose *E. faecalis* (EF 5x10⁷ CFU) with *M. luteus* (ML 2x10⁸ CFU) into mice: liver CFU (n = 10 per group, median value shown, two-tailed Mann-Whitney test), ****p<0.0001 **G** Summary heat-map of alternative pathogens tested for ability to be augmented, with addition of PGN or *M. luteus*, showing significant changes in liver CFUs and weight change. In all cases liver CFUs were enumerated at 3 days post-infection.

S2H Fig). These data suggest that augmented infections which result in increased pathogen numbers from the inoculum may be specific to *S. aureus* and a facet of its particular interaction mechanism with the host.

## Augmentation requires spatial and temporal co-localisation of *S. aureus* and augmenting material

Macrophages and ROS are implicated in augmentation [12], suggesting that augmenting material influences *S. aureus* infection within the phagocyte. To test whether augmentation in the murine sepsis model requires concomitant inoculation of augmentor/pathogen, PGN was injected at a range of timepoints before and after *S. aureus* infection.

PGN was injected into mice at 24, 6 or 1 hours before, or 0, 6, 24 or 48 hours after *S. aureus* infection (Fig 2A–2F). *S. aureus* liver bacterial numbers were significantly increased when PGN was co-injected, but not when PGN was injected at all time-points before (Fig 2A–2C) or after (Fig 2D–2F) *S. aureus*, suggesting that co-administration is required. No change in weight or kidney bacteria was observed for PGN injected before (Figs 2C and S3A, S3B), while, for PGN injected after *S. aureus*, no change in kidney bacteria was observed, but significant reductions in mouse weight loss were observed at 6 and 24 hours (Figs 2F and S3C, S3D). Together these data demonstrate that to increase *S. aureus* pathogenesis, augmenting material must be present concomitantly with *S. aureus*.

The requirement for concomitant administration of *S. aureus* and augmenting material suggested that they are likely co-phagocytosed. To examine this, the amount of augmenting

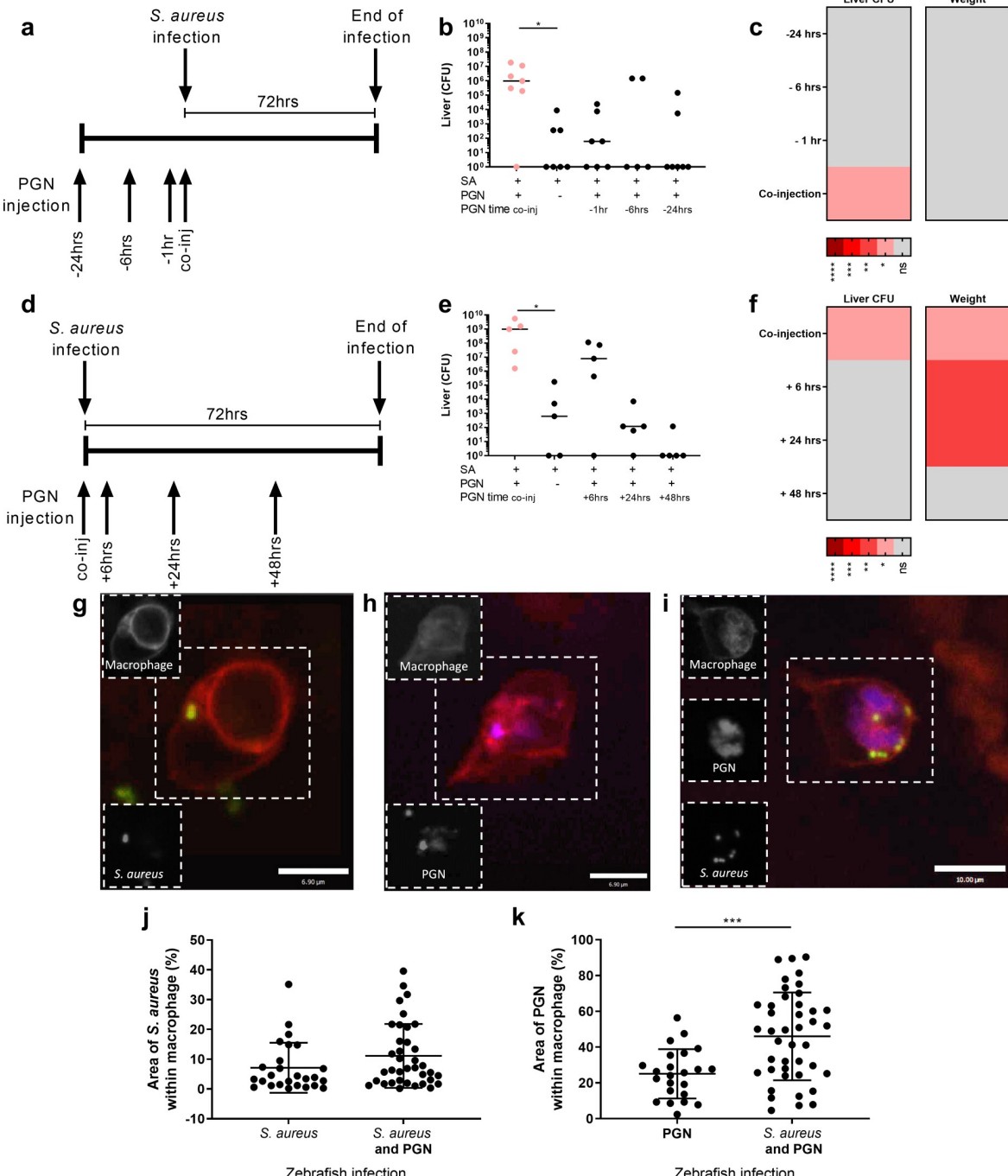

**Fig 2. Augmenting material must be present at the same time and location as *S. aureus* to enhance infection. A-C** Mice were intravenously injected with 500 μg *M. luteus* PGN 24 hours, 6 hours or 1 hour before infection with 1x10[6] *S. aureus*, or at the same time as *S. aureus*, or with *S. aureus* alone **A** Diagram of experimental protocol **B** liver CFUs, enumerated at 72 hpi (n = 7 per group, median value shown, Kruskal-Wallis test with Dunn's post-test), *p*<0.05 **C** Summary heat-map for augmenting ability of PGN added before *S. aureus* infection, showing significant changes in liver CFUs and weight change **D-F** Mice were intravenously injected with 500 μg *M. luteus* PGN 48 hours, 24 hours or 6 hours after infection with 1x10[6] *S. aureus*, or at the same time as *S. aureus*, or with *S. aureus* alone **D** Diagram of experimental protocol **E** liver CFUs, enumerated at 72 hpi (n = 5 per group, median value shown, Kruskal-Wallis test with Dunn's post-test) *p*<0.05 **F** Summary heat-map for augmenting ability of PGN added after *S. aureus* infection, showing significant changes in liver CFUs and weight change **G-K** Zebrafish larvae injected with 400 CFU *S. aureus*, 5 ng of *M. luteus* PGN, or both. The larvae have fluorescent macrophages (red) and were injected with fluorescent *S. aureus* (green) and/or fluorescently labelled *M. luteus* PGN (blue) **G-I** Images of infected larvae at 2 hpi showing macrophages containing *S. aureus*, scale 6.9 μm, greyscale insets depict location of fluorescence signal within the hatched box of the main image, for ease of visualisation (**G**), *M. luteus* peptidoglycan scale 6.9 μm (**H**), or both scale

10 μm (**I**), **J** Area of macrophage taken up by *S. aureus* at 2 hpi (n = 3, 14–21 larvae per group, unpaired t-test) **K** Area of macrophage taken up by *M. luteus* PGN at 2hpi (n = 3, 11–21 larvae per group, two-tailed unpaired t-test, \*\*\**p*<0.0004).

material within individual phagocytes *in vivo* was examined. A zebrafish transgenic line with fluorescent macrophages was used: *Tg(mpeg:mCherry.CAAX)sh378* [35]. Augmentation has previously been shown to occur during systemic infection of zebrafish larvae [12]. Larvae were infected with GFP fluorescent *S. aureus* and/or fluorescently stained PGN. Macrophages phagocytosed injected material in each individually injected group (Fig 2G and 2H) and *S. aureus* and PGN were co-localised when present within the same macrophage (Figs 2I and S3E). Macrophages were imaged and the area of phagocytosed fluorescent materials was quantified using Fiji. The area taken up by *S. aureus* within individual macrophages was not altered when PGN was present (Fig 2J). However, the area of PGN was significantly increased in the presence of *S. aureus*, in comparison to PGN injected alone (Fig 2K). Thus, augmentation does not alter the level of *S. aureus* phagocytosis *in vivo*, however, it appears that macrophages which engulf *S. aureus* also phagocytose more augmenting material.

## Augmenting material protects *S. aureus* from ROS *in vitro*

Since augmentation does not occur in the absence of NOX2, and Kupffer cells have reduced ROS levels in augmented *S. aureus* infection [12], we hypothesised that inactivation of ROS by augmenting material could be the mechanism by which *S. aureus* survival is enhanced with an ensuing increase in pathogenesis. We therefore tested whether augmenting material protects *S. aureus* from specific ROS and RNS *in vitro*, using $H_2O_2$, sodium hypochlorite (a source of HOCl), peroxynitrite, and methyl viologen (a source of superoxide) (Fig 3A). *S. aureus* survival in liquid culture *in vitro* was measured following ROS exposure, with or without *M. luteus* PGN. Exposure to each ROS led to a significant reduction in *S. aureus* numbers, while addition of PGN significantly increased *S. aureus* survival in the presence of $H_2O_2$, HOCl and peroxynitrite, but not methyl viologen (Figs 3B–3D and S4A).

We have previously shown that live bacteria augment *S. aureus* infection [12], as such, we hypothesised that *M. luteus* would promote *S. aureus* survival in the presence of ROS. *M. luteus* was used at 100 times the concentration of *S. aureus*. Addition of live *M. luteus* led to significantly increased survival of *S. aureus* after $H_2O_2$, HOCl and peroxynitrite treatments, but not methyl viologen (S4B–S4E Fig). It was possible that live *M. luteus* was mediating augmentation via production of ROS defence enzymes, such as catalase. Addition of HK *M. luteus* increased *S. aureus* survival when exposed to HOCl but not $H_2O_2$ and peroxynitrite (Figs 3E and S4B–S4D). To determine if the lack of effectiveness of HK *M. luteus* was due to the availability of ROS active moieties the ratio of augmenting material was raised (ratio of 1:2500), which significantly increased *S. aureus* survival following exposure to $H_2O_2$ or peroxynitrite (Fig 3F and 3G). Thus, both live and HK *M. luteus* can protect *S. aureus* from ROS. It appears that $H_2O_2$ or peroxynitrite are effectively deactivated by enzymes present in live *M. luteus*; although these enzymes promote augmentation they are not required.

Thus, augmenting material may act as a buffer to react with, and therefore detoxify, ROS. If this is so, pre-treatment of augmenting material with ROS would diminish its effect. To test this, live *M. luteus* were pre-treated with ROS prior to inclusion in the *in vitro* liquid culture assay. HOCl pre-treated *M. luteus* showed a clear loss of protective ability, with no surviving bacteria, in comparison to live or HK *M. luteus* (Fig 3E). For both $H_2O_2$ and peroxynitrite, the level of *S. aureus* survival with addition of ROS-treated *M. luteus* was ~10–20%, whereas with HK *M. luteus* this was ~100% (Fig 3F and 3G). This indicates that augmenting material has a finite capacity to react with ROS and, in so doing, loses its ability to protect *S. aureus*.

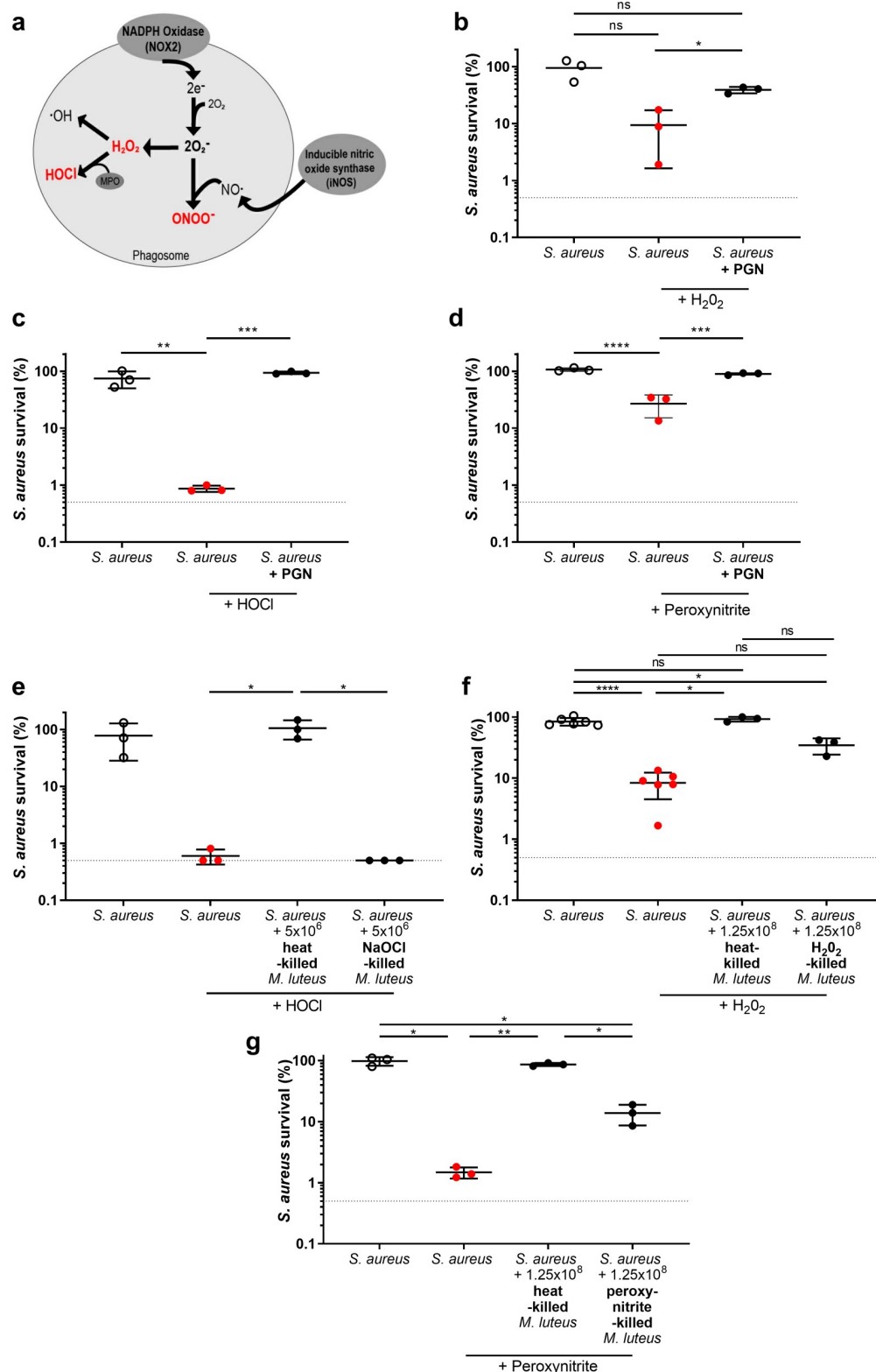

**Fig 3. Augmenting material protects *S. aureus* from ROS *in vitro*. A** Reactive oxygen species generated in the oxidative burst, ROS highlighted in red are examined **B-D** Following ROS exposure, percentage survival of *S. aureus* alone ($5 \times 10^4$ CFU/mL), or with *M. luteus* PGN (1.25 mg) **B** hydrogen peroxide (n = 3, error bars show mean +/- SD, one-way ANOVA test with Tukey's post hoc test) *$p < 0.05$ **C** sodium hypochlorite (n = 3, error bars show mean +/-

SD, one-way ANOVA test with Tukey's post hoc test), $**p<0.01$; $***p<0.005$ **D** peroxynitrite (n = 3, error bars show mean +/- SD, one-way ANOVA test with Tukey's post hoc test), $***p<0.005$; $****p<0.0001$ **E** Following sodium hypochlorite exposure, percentage survival of *S. aureus* alone ($5x10^4$ CFU/mL), with heat-killed *M. luteus* (equivalent of $5x10^6$ CFU/mL) or ROS killed *M. luteus* (equivalent of $5x10^6$ CFU/mL) (n = 3, error bars show mean +/- SD, one-way ANOVA test with Tukey's post hoc test), $*p<0.05$ **F-G** Following ROS exposure, percentage survival of *S. aureus* alone ($5x10^4$ CFU/mL), with heat-killed *M. luteus* (equivalent of $1.25x10^8$ CFU/mL), or ROS killed *M. luteus* (equivalent of $1.25x10^8$ CFU/mL) **F** hydrogen peroxide (n = 3, +/-SD, one-way ANOVA test with Tukey's post hoc test), $*p<0.05$; $****p<0.0001$ **G** peroxynitrite (n = 3, +/-SD one-way ANOVA test with Tukey's post hoc test), $*p<0.05$; $**p<0.003$.

## Augmenting material restores virulence to ROS susceptible *S. aureus in vitro* and *in vivo*

*S. aureus* mutants lacking oxidative stress resistance mechanisms are susceptible to ROS and attenuated in pathogenesis [20,21]. We used a *S. aureus katA ahpC* to map which ROS resistance mechanisms are important for *S. aureus* survival and to test the ability of augmenting material to rescue this strain. The place of the ROS resistance enzymes investigated here in detoxifying the oxidative burst is shown in Fig 4A. *S. aureus katA ahpC* would be expected to have a reduced ability to detoxify $H_2O_2$, organic peroxides and peroxynitrite, and is more sensitive to peroxides *in vitro* [21]. *S. aureus katA ahpC* was protected from $H_2O_2$ by live *M. luteus*, but not by HK or $H_2O_2$-treated cells (Fig 4B), as were individual *katA* or *ahpC* mutants (S5A and S5B Fig). When *katA ahpC* was exposed to HOCl, *katA ahpC* survival was significantly increased from ~0.2% to ~100% with the addition of *M. luteus* and to ~68% with HK *M. luteus*, but not ROS-treated *M. luteus* which remained at ~4.4% survival (S5C Fig). The role of ROS resistance was then tested *in vivo* using the murine sepsis model. The *katA*, *ahpC* and *katA ahpC* strains were attenuated, with significantly fewer liver bacteria recovered for *ahpC* and *katA ahpC*, and kidney bacteria for *katA ahpC* and *katA*, with all strains causing significantly reduced weight loss in comparison to wild-type (S5D–S5I Fig).

To test whether ROS-susceptible *S. aureus* could be augmented *in vivo*, a low dose ($1x10^6$ CFU) of *katA ahpC* was injected with or without HK *M. luteus* ($1x10^8$ CFU). In the presence of augmenting material, the *katA ahpC* strain had an exceptionally large and significant increase in liver bacterial numbers from 0 CFU to ~$3x10^7$ CFU, levels seen in augmented wild-type *S. aureus* infections, but no significant change in weight loss or kidney bacterial numbers (Figs 4C and S5J, S5K). Thus *S. aureus katA ahpC* can not only be augmented but also this leads to loss of attenuation in the liver. This further supports the assertion that augmentation occurs in the liver and is associated with the ability of the bacteria to survive ROS, as well as that augmentation also occurs during the initiation of infection. ROS resistance is additionally required for later infection stages, as *S. aureus katA ahpC* in the presence of augmenting material does not recover to parental bacterial numbers in the kidney.

To examine the role of HOCl in augmented *S. aureus* murine infection, MPO knock-out mice (MPO$^{-/-}$) were infected alongside wild-type controls. Interestingly, we observed a significant increase in liver bacterial numbers in MPO$^{-/-}$ mice in comparison to wild-type mice (Fig 4D), demonstrating that MPO$^{-/-}$ mice were more susceptible to *S. aureus* infection. This suggests that MPO activity is a crucial component of the host defence in this model. Augmented infections for both wild-type and MPO$^{-/-}$ mice had significantly more liver bacterial numbers and increased weight loss than respective non-augmented infections (Figs 4D and S5L). Therefore, loss of MPO did not inhibit the ability of augmenting material to enhance *S. aureus* infection, suggesting that HOCl amelioration, at least alone, is not sufficient for augmentation *in vivo*.

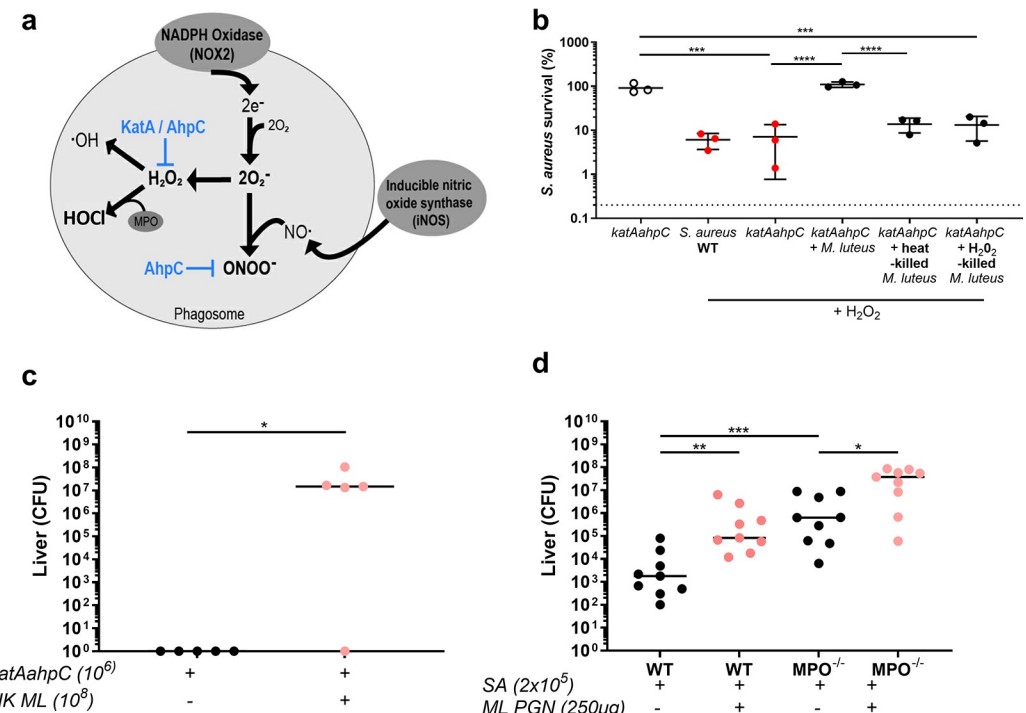

**Fig 4. *S. aureus* strains susceptible to ROS are augmented *in vitro* and *in vivo*. A** Reactive oxygen species generated in the oxidative burst, highlighting where KatA and AhpC action occurs **B** Following hydrogen peroxide exposure, percentage survival of *S. aureus katAahpC* mutants alone (5x10⁴ CFU/mL), or with live *M. luteus* (5x10⁶ CFU/mL), heat-killed *M. luteus* (equivalent of 5x10⁶ CFU/mL) or ROS killed *M. luteus* (equivalent of 5x10⁶ CFU/mL), (n = 3, error bars show mean +/- SD, one-way ANOVA test with Tukey's post hoc test), $^{***}p<0.0002$; $^{****}p<0.0001$ **C** Co-injection of low dose *S. aureus katAahpC* (1x10⁶ CFU) with heat-killed *M. luteus* (equivalent of 1x10⁸ CFU) into mice: liver CFU, enumerated at 3 days post-infection (n = 5 per group, median value shown, two-tailed Mann-Whitney test), $^{*}p<0.05$ **D** Co-injection of low dose *S. aureus* (SA 2x10⁵ CFU) with *M. luteus* PGN (ML PGN 500 μg) into wild-type (WT) control mice or MPO⁻/⁻ mice: liver CFU, enumerated at 3 days post-infection (n = 9 per group, median value shown, individual two-tailed Mann-Whitney tests), $^{*}p<0.05$, $^{**}p<0.002$, $^{***}p<0.0006$.

## Augmenting material protects *S. aureus* by inactivating ROS in macrophages

Augmentation occurs at the initiation of infection by circumventing the deleterious effects of ROS *in vivo*. To determine how these manifest at the cellular level, we used a murine macrophage cell line. Time-lapse imaging of RAW264.7 cells infected with fluorescent *S. aureus* were used to examine whether bacteria surviving within macrophages may represent the source of the microabcesses that occur as a product of augmentation [12]. In the presence of augmenting material, intracellular *S. aureus* survival and growth were observed within individual macrophages, which eventually led to host cell death and formation of large extracellular accumulations of bacteria, referred to here as bacterial masses (Fig 5A). Using a high-throughput assay to examine *S. aureus* mass formation, RAW264.7 cells were infected with *S. aureus* with or without HK *M. luteus* at a ratio of 1:10, a lower ratio than was used in the preceding *in vitro* and *in vivo* work, to limit cell toxicity. Despite this, the number of masses was significantly increased in the augmented group in comparison to *S. aureus* infection alone (Fig 5B). We next examined the ratio of augmenting material to *S. aureus*, using an augmenting material ratio of 10, 5, 2.5, 0.5 and 0.05 to *S. aureus*, with increased numbers of masses forming in the presence of higher concentrations of augmenting material (Fig 5C). Higher *S. aureus* levels also led to increased mass formation (Fig 5D). These data demonstrate dose-dependent

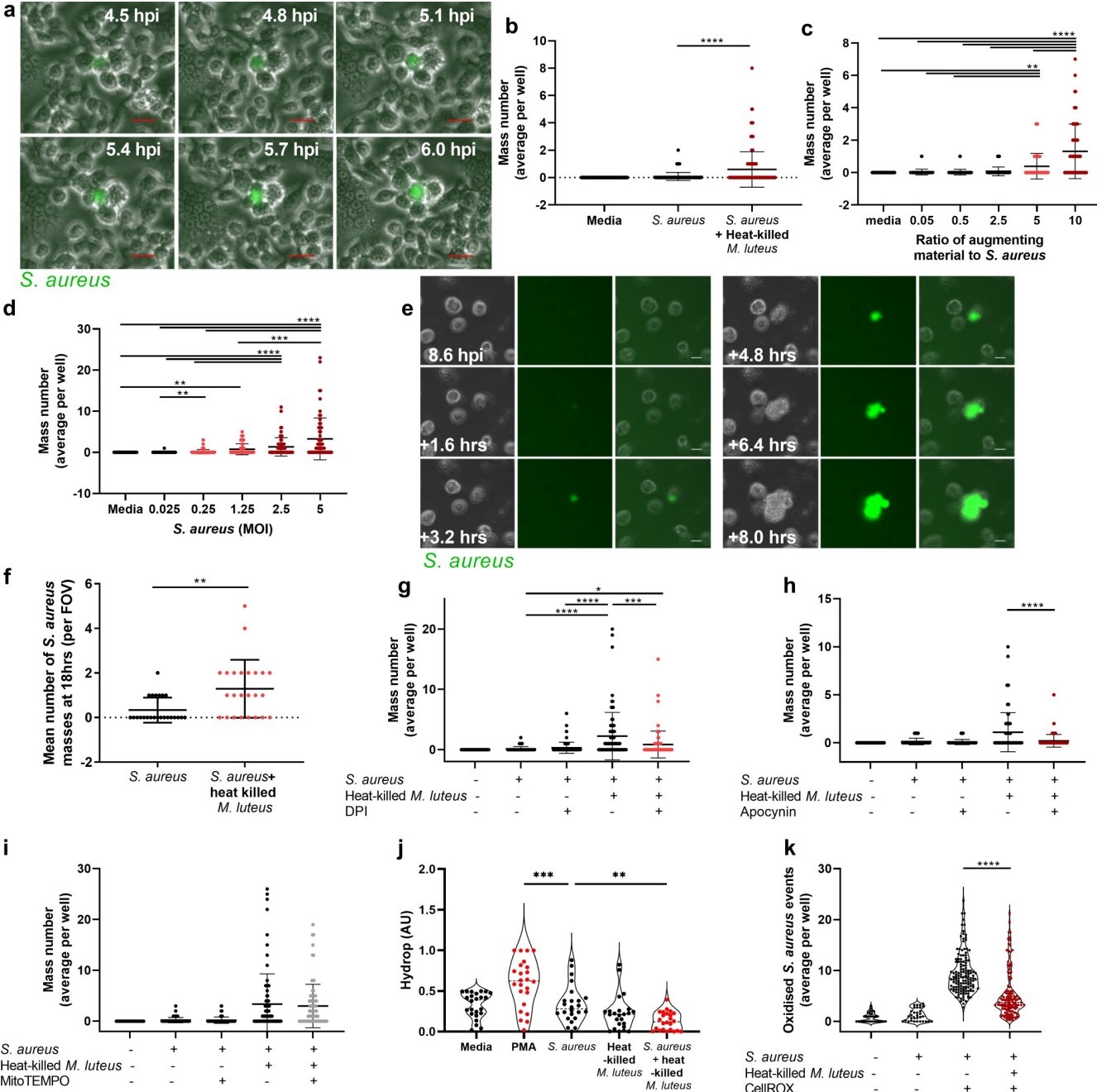

**Fig 5. Augmenting material protects *S. aureus* from reactive oxygen species in macrophages. A** Images of GFP-*S. aureus* mass formation within RAW264.7 cells, scale 20 μm **B** RAW264.7 cells infected with GFP *S. aureus* (MOI 5) with or without heat-killed *M. luteus* (MOI 50), (n = 4), ****$p<0.0001$ **C** RAW264.7 cells infected with GFP *S. aureus* (MOI 5) with or without heat-killed *M. luteus* (ratio to *S. aureus*, 10, 5, 2.5, 0.5 or media control), (n = 4), **$p<0.008$; ****$p<0.0001$ **D** RAW264.7 cells infected with GFP *S. aureus* (MOI 5, 2.5, 1.25, 0.25, 0.025 or media control) with or without of heat-killed *M. luteus* (MOI 50), (n = 4), **$p<0.003$; ***$p<0.0008$; ****$p<0.0001$ **E-F** MDMs infected with GFP *S. aureus* (MOI 5) with or without heat-killed *M. luteus* (MOI 50) **E** images of GFP *S. aureus* mass formation within human MDMs, scale 20 μm **F** number of *S. aureus* masses observed (n = 3), **$p<0.003$ **G** RAW264.7 cells infected with GFP *S. aureus* (MOI 5) in the presence or absence of heat-killed *M. luteus* (MOI 50), either with or without DPI (2 μM), (n = 4), *$p<0.05$;***$p<0.0004$; ****$p<0.0001$ **H** RAW264.7 cells infected with GFP *S. aureus* (MOI 5) in the presence or absence of heat-killed *M. luteus* (MOI 50), either with or without apocynin (500 μM), (n = 4), ****$p<0.0001$ **I** RAW264.7 cells infected with GFP *S. aureus* (MOI 5) in the presence or absence of heat-killed *M. luteus* (MOI 50), either with or without mitoTEMPO (1 μM), (n = 4, non-significant) **J** RAW264.7 cells infected with GFP *S. aureus* (MOI 5) with or without heat-killed *M. luteus* (MOI 50) with Hydrop used to visualise hydrogen peroxide (n = 4, violin plot with median values shown), **$p<0.007$; ***$p<0.0004$ **K** RAW264.7 cells infected with CellROX-stained GFP *S. aureus* (MOI 50) to visualise intracellular oxidation in the presence or absence of heat-killed *M. luteus* (MOI 50), (n = 4, violin plot with median values shown), ****$p<0.0001$. In panels B, F, H, I and K, a two-tailed Mann Whitney test was used, in panels C, D, G, and J, a Kruskal-Wallis test with Dunn's post hoc test was used. Where used, error bars show mean +/- SD.

augmentation by HK bacteria of *S. aureus* survival and proliferation within macrophages. Finally, we used human monocyte-derived macrophages (MDMs) in the time-lapse *S. aureus* mass formation assay. *S. aureus* was able to survive, proliferate and escape from MDMs (Fig 5E). Similarly, *S. aureus* mass formation from MDMs was significantly increased in the presence of augmenting material (Figs 5F and S6A). Augmenting material therefore increases the capacity of *S. aureus* to overwhelm human macrophages.

To examine the importance of ROS production, the mass formation assay was evaluated following treatment with NOX2 inhibitors DPI or apocynin, using concentrations which did not inhibit *S. aureus* growth (S6B and S6C Fig). Addition of DPI or apocynin significantly reduced the level of augmentation compared to the untreated controls (Fig 5G and 5H), but treatment with a specific scavenger of mitochondrial superoxide (mitoTEMPO) did not (Fig 5I). This confirms that ROS production, specifically in phagosomes, is important for augmentation of *S. aureus* infection within macrophages. Our *in vitro* assays showed that augmenting material protects *S. aureus* from ROS. Therefore, we used a specific fluorescent probe, Hydrop, to examine levels of $H_2O_2$ within infected RAW264.7 macrophages. The Hydrop assay showed significantly reduced $H_2O_2$ levels in RAW264.7 cells infected with *S. aureus* and HK *M. luteus*, in comparison to *S. aureus* alone (Figs 5J and S6D). To further examine how augmentation affects oxidation of the bacteria, RAW264.7 macrophages were infected with GFP *S. aureus* stained with CellROX, a dye which becomes fluorescent when oxidised by ROS. There were significantly more oxidised *S. aureus* events in macrophages infected with *S. aureus* alone than those infected with *S. aureus* alongside HK *M. luteus* (Figs 5K and S6E). Together these data demonstrate ROS levels are reduced in the presence of augmenting material, suggesting this material acts to inactivate ROS.

## Can augmentation be exploited for vaccine development?

*S. aureus* vaccine development has been unsuccessful, in part due to animal models not being representative of human disease and high variability in infection outcome [36]. Natural human infection with *S. aureus* emerges from a polymicrobial environment. We therefore tested if the augmented infection model (where augmenting material represents polymicrobial species) might provide a suitable framework for vaccine development.

The test vaccine consisted of 1 μg ClfA, 50 μg CpG and 1% w/v Alhydrogel an aluminium based adjuvant, components which have been used previously [37–39]. Mice were vaccinated on days 0, 14 and 21 before *S. aureus* infection on day 28, with blood drawn before and after vaccinations (Fig 6A). The vaccine was tested for efficacy, alongside PBS control injections, in three *S. aureus* infection scenarios; low dose ($1x10^6$ *S. aureus*), high dose ($1x10^7$ *S. aureus*) and augmented low dose ($1x10^6$ *S. aureus* +/- *M. luteus* PGN), in 2 independent experiments (Figs 6 and S7). Low dose infection caused low numbers (an average of 38 CFU) of bacteria in the liver, with more observed in the high dose infection (an average of $1.43x10^6$ CFU), and as expected, the augmentation groups had very high liver bacterial numbers (an average of $4.45x10^7$ CFU) (Fig 6B). Interestingly, vaccination reduced *S. aureus* liver and kidney bacterial numbers only for the augmentation groups (Figs 6B and S7A, S7B). The second independent experiment also showed that the vaccine was only effective in reducing *S. aureus* pathogenesis in the augmented scenario (S7C–S7F Fig). These data suggest that using augmentation to examine vaccine efficacy may be a useful strategy, as it mimics natural infection.

## Discussion

*S. aureus* is an insidious pathogen made more concerning due to the spread of antimicrobial resistance and the lack of an available vaccine. Understanding infection dynamics provides a

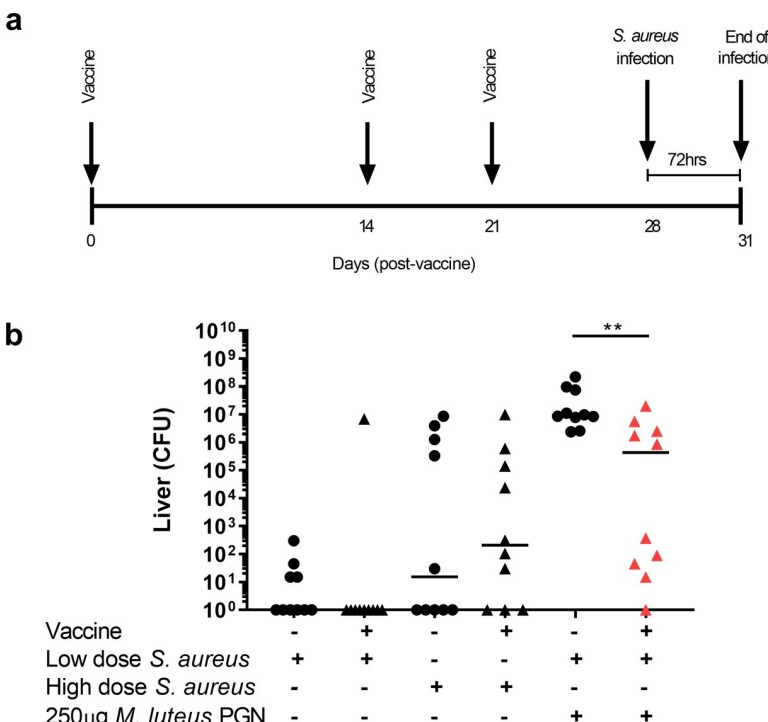

**Fig 6. Vaccination reduces augmented infection. A-B** Mice were vaccinated subcutaneously on day 0, 14 and 21 with vaccine (1 µg ClfA, 50 µg CpG and 1% w/v Alhydrogel, triangles) or PBS control (circles). Day 28 post-vaccination mice were intravenously injected with low dose *S. aureus* ($1 \times 10^6$ CFU) high dose *S. aureus* ($1 \times 10^7$ CFU), with both low dose *S. aureus* and 250 µg *M. luteus* PGN **A** diagram of experimental protocol **B** liver CFUs, enumerated 72 hpi (n = 10 per group, median value shown, two-tailed Mann-Whitney test), $^{**}p < 0.002$.

route to the identification of disease breakpoints where interventions might be most effective. An effective vaccine should be able to prevent disease establishment, and so understanding the status of the pathogen at this infection initiation stage is crucial. All pathogens exist within a polymicrobial environment from which they emerge to cause disease. *S. aureus* lives as a human commensal, primarily in the nares where even in this niche it forms only a small proportion of the microbiome [1–3]. Thus, all *S. aureus* infections are initiated from an inoculum that is mostly not the pathogen. With this backdrop, we have identified the augmentation phenomenon, where human-skin commensals or derivatives enhance *S. aureus* pathogenesis, acting at the level of initial macrophage interaction [12]. The amount of material required to augment *S. aureus* infection is comparable to the number of bacteria located on human skin or vascular catheters [40,41]. Here we find that augmentation is not specific to *S. aureus* as it occurs with other opportunist pathogens. Both *E. coli* and *E. faecalis*, which survive intracellularly within macrophages [42,43], benefitted from augmenting material. However, in both cases augmentation was evidenced by reduced clearance rather than an increase in pathogen load, suggesting the increase in pathogenesis resulting in increased bacterial burden may be peculiar to *S. aureus*. We also show *S. aureus* disease can be augmented by a range of particulate materials from whole bacteria to fungal cell walls, suggesting that augmentation is not mediated by a response to specific components. This is supported by our previous work that demonstrated augmentation not to require any of the major host response pathways such as NOD1 and NOD2 [12]. The hypothesis that augmentation occurs at the initiation of infection was further supported by the requirement for co-inoculation of *S. aureus* and augmenting material.

Augmentation has a profound effect on *S. aureus* disease, resulting in the ability to reduce the required inoculum by 1000-fold to cause systemic disease in the murine sepsis model [12]. To determine what the molecular mechanism might be, we homed in on those events which occur within the macrophage after phagocytosis, where ROS production is known to be required for augmentation [12]. *In vitro*, augmenting material protected *S. aureus* from $H_2O_2$, HOCl and peroxynitrite, suggesting that augmenting material reacts with ROS acting as a buffer, allowing continued *S. aureus* survival. Augmenting material showed a variable, protective capacity against different ROS. As an example, low dose HK *M. luteus* (equivalent CFU $5x10^6$) protected WT *S. aureus* from killing by HOCl but not the other ROS tested (Figs 3E and S4C, S4D). Conversely, PGN (25 mg/ml) was able to protect *S. aureus* from $H_2O_2$, peroxynitrite and HOCl (Fig 3B–3D). Use of a higher dose of HK *M. luteus* (equivalent CFU $1.25x10^8$), which was comparable to the number of live cells from which the PGN was derived gave protection to all 3 ROS demonstrating parity (Fig 3F and 3G). Live *M. luteus*, at a concentration of $10^6$ CFU, was able to protect *S. aureus* from ROS killing by HOCl, $H_2O_2$ and peroxynitrite, which is likely due in part to *M. luteus* ROS resistance enzymes, such as catalase. A variety of biological entities present on augmenting material hold the potential to react with ROS resulting in, for example, oxidation or chlorination [44,45]. Furthermore, pre-treatment of augmenting material with ROS inhibited its protective ability, defining a finite capacity for ROS detoxification. The *in vitro* data was obtained in an environment very different from that experienced by the bacteria inside phagocytes, let alone *in vivo*, therefore it was important to make analyses in these more complex milieu. There are a range of ROS, all ultimately originating from superoxide as a product of NADPH oxidase, but which are directly involved in *S. aureus* killing in macrophages is unknown [26]. *S. aureus katA ahpC* is susceptible to $H_2O_2$ *in vitro* and is attenuated *in vivo*. Interestingly, augmentation had a dramatic effect on *S. aureus katA ahpC* pathogenesis resulting in extremely boosted virulence, to a level compatible to its parent. This embeds the role of ROS resistance at the very earliest stages of disease in order to pass the initial threshold of infection establishment. $H_2O_2$ is produced early during oxidative burst [16,17,46] and may therefore constitute a key ROS in controlling *S. aureus*. We also found that augmenting material protects *S. aureus* from HOCl and that pre-treatment of the augmentor with this ROS abrogated its protective effect. HOCl is derived from $H_2O_2$ by MPO within macrophages but at a proposed lower level than in neutrophils [47], leading to a higher $H_2O_2$ concentration in macrophages [48]. Nevertheless, here we demonstrated that MPO is an important host defence enzyme *in vivo*, where its loss resulted in increased bacterial load in the liver highlighting HOCl as an important ROS in the control of infection. Lack of MPO did not prevent augmentation, in contrast to the loss of NOX2 activity in mice which did [12]. It is therefore likely that augmenting material acts as a sink for ROS in general thereby protecting *S. aureus* and allowing it to survive this crucial phase in host innate defences.

The effect of augmenting material is to allow *S. aureus* to survive the ROS assault in the macrophage. Inhibition of ROS in the absence of augmenting material did not greatly enhance *S. aureus* mass formation in isolated macrophages, possibly indicating that other killing mechanisms, of which there are a variety [26], may compensate *in vitro*. When *S. aureus* infection is augmented, absorption of ROS by augmenting material may prevent further maturation of the phagosome and thus activation of downstream bactericidal mechanisms. However, the importance of host ROS in controlling *S. aureus* infection real-life infections is clearly demonstrated with increased *S. aureus* pathogenicity in MPO (or NOX2 [12]) deficient mice, as well as the attenuation of *S. aureus katA ahpC* infection *in vivo*. A model for the molecular mechanism of augmentation is shown in S8 Fig, where phagocytosis of a threshold number of *S. aureus* leads to activation of ROS production and bacterial killing. Augmentation results in a bolus of phagocytosed material in addition to the *S. aureus* that acts to detoxify ROS and so increase

the chance of bacterial survival, subsequent proliferation and lysis of the phagocyte, releasing a cluster of bacteria able to further multiply to form a microabscess. It is these microabscesses that can then go onto seed other sites in the host leading to a systemic and potentially fatal infection. Augmentation may act, therefore, to increase chances of infection spread by expanding the number of macrophages that are ineffective at controlling the initial infective dose. As the initiation of human infection will come from a polymicrobial environment, augmentation provides a framework to test prophylactic regimen. Indeed, under an augmentation scenario, an experimental *S. aureus* vaccine reduced bacterial burden. Understanding infection dynamics and the interplay between pathogen, host and other organisms is beginning to give insight into disease progression, and how novel interventions to sway the outcome in the favour for the host may be derived.

# Materials and methods

## Ethics statement

Animal work in the UK was performed in accordance with the Animal (Scientific Procedures) Act 1986. At the University of Sheffield, work was completed under project licences P3BFD6DB9 and PPL 40/3699 for murine work, or P1A4A7A5E for zebrafish work, with ethical approval from the University of Sheffield Local Ethical Review Panel. At Imperial College London, work was conducted under licence P4C824899 with approval from the Imperial ethical review board. At INRAE, animal work was approved by the local ethics committee (COMETHEA or "Comité d'Ethique en Expérimentation Animale", Centre de Recherche Ile de France—Jouy en Josas–Antony) under the registration numbers 15_08, and by the French Ministry of Higher Education and Research APAFIS #480-2015041518048149v1, where all animal experiments were performed in accordance with European directive 2010/63/EU. Animal experiments in Calgary were approved by the University of Calgary Animal Care Committee and were in compliance with the Canadian Council for Animal Care Guidelines (protocol nr. AC16-0148). MDMs were derived, with informed consent, from the blood of healthy volunteers, in accordance with guidelines from the South Sheffield Research Ethics Committee (07/Q2305/7).

## Animal husbandry

Mice were housed in designated animal facilities in standard husbandry conditions. BALB/c female mice aged 6–9 weeks (Charles River Laboratories) for all animal work completed at the University of Sheffield. At the University of Calgary female and male MPO$^{-/-}$ Mice and C57 wild-type controls aged 8–11 weeks (Jackson Laboratory) were used. Adult zebrafish were maintained according to standard protocols in UK home office approved facilities, at the Bateson centre Aquaria at the University of Sheffield. Embryos less than 5.2 days post-fertilisation (dpf) of *Tg(mpeg1:mCherryCAAX)sh378* [35] were used.

## Murine models

The mouse sepsis model was completed by injecting BALB/c mice with a pathogen (*S. aureus*, *E. faecalis*, *S. pneumoniae* or *P. aeruginosa*), augmenting material (*M. luteus*, PGN, HK *M. luteus*, HK *C. neoformans*, *E. coli*, *R. mucosa* or *S. cerevisiae*), or mixture of a pathogen and an augmentor. These were injected into the tail vein in a volume of 100 μl. Individual experimental figure legends show the CFU and/or amount of augmenting material used. Bacteria were prepared for injection as previously established [49], and serial dilutions of the inoculum were plated to confirm CFUs injected into mice. Mice were monitored daily for health and weight,

and were euthanised at the experimental end point of 48 or usually 72 hours post-infection (hpi). Liver and kidney CFU were calculated as previously established [12].

For murine intramuscular injections, mice were challenged intramuscularly with *S. pyogenes*, with or without 500 μg *M. luteus* PGN in 50 μl PBS and quantitative endpoints compared at 24 hpi. Mice were euthanised and thigh muscle dissected and then homogenised. Bacterial CFU counts were determined by plating of homogenised tissue and blood samples onto the specified agar, with or without dilution in PBS, as appropriate.

In the subcutaneous mouse injections used in the vaccine experiments, mice were scruffed and injected subcutaneously into the scruff with 100 μl of vaccine or PBS control.

For murine blood sampling, mice were warmed to 37 °C to promote tail vein dilation, a small cut was made into the tail vein allowing a small volume of blood to be collected.

## Zebrafish infection model

For zebrafish infection and imaging, the established zebrafish infection model was followed [50]. Immediately prior to injection, PGN was stained with Alexa Fluo 405 NHS Ester (Fisher) following established protocols [51]. 2 dpf larvae were anesthetized with tricaine and injected with 1 nl containing 400 CFU of GFP-*S. aureus*, 5 ng of stained PGN, or both, into the yolk sac circulation valley. Larvae were then recovered, before being mounted in 0.8% low melt agarose (Affymetrix, 32830) in glass-bottom microwell dishes (MatTek, P35 G-1.5–14 C). An Ultra-VIEW VoX spinning disk confocal microscope (Perkin Elmer, Cambridge, UK) was used for imaging larvae at 2 hpi, where 405 nm, 445 nm, 488 nm, 514 nm, 561 nm and 640 nm lasers were available for excitation and a 40x oil objective (UplanSApo 40x oil [NA 1.3] was used. Imaging of macrophages were obtained in the caudal hematopoietic tissue (CHT). Analysis was carried out using Fiji (ImageJ) to measure the area of macrophages, and *S. aureus* and PGN within macrophages. 47 macrophages from 21 zebrafish larvae which had been co-injected with *S. aureus* and PGN were assessed for the presence of phagocytosed *S. aureus*, PGN or both.

## Bacterial strains and culture

Microbial strains used in this study are listed in Table 1.

## Transductions

*S. aureus* NewHG *katA*, *ahpC* and *katA ahpC* strains were created using established transduction protocols [21] using ϕ11, from existing *S. aureus* SH1000 *katA* (KS100) [21] or *ahpC* (KC041) [21] strains. Presence of mutations in the newly generated NewHG strains were confirmed with PCR. The NewHG-mCherry (SJF4439) was created following the same transduction protocols [21] using an existing SH1000-mCherry strain [13].

## Peptidoglycan and capsule preparation

*M. luteus* or *S. aureus* PGN used throughout this study was prepared and purified using established protocols [55]. *C. neoformans* capsule was isolated after growing cultures for 7 days at 28˚C at 180 rpm. The culture was autoclaved, cells harvested at 6000 *g*, and supernatant collected. Ice-cold ethanol was added to precipitate the cryptococcal capsule, which was collected after centrifugation again at 6000 *g*.

## Vaccine preparation

Vaccine formulations were made in 1x endotoxin free PBS. Vaccines were used at a final concentration of 1 μg/dose Clumping factor A (ClfA, recombinant, endotoxin purified, mass-

**Table 1. Bacterial and fungal strains used.**

| Species | Strain | Description | Culture conditions | Reference or source |
|---|---|---|---|---|
| *Staphylococcus aureus* | NewHG (SJF3680) | NewHG *lysA*::pGM072(Kan$^R$) *lysA*+ | Tryptic Soy Broth (TSB, Kanamycin 50 μg/mL), 37˚C | [11] |
| *Staphylococcus aureus* | NewHG-GFP (SJF4620) | *geh*::Pma1M-GFP | TSB (Kanamycin 50 μg/mL), 37˚C | [13] |
| *Staphylococcus aureus* | NewHG-mCherry (SJF4439) | pMV158-mCherry *lysA*::kan *lysA*+ | TSB (Kanamycin 50 μg/mL and Tetracycline 5 μg/mL), 37˚C | This study |
| *Staphylococcus aureus* | NewHG *katAahpC* (SJF5252) | *katA*::Tn917 and *ahpC*::tet | TSB (Tetracycline 5 μg/mL, Erythromycin 5 μg/mL), 37˚C | This study |
| *Staphylococcus aureus* | NewHG *katA* (SJF5251) | *katA*::Tn917 | TSB (Erythromycin 5 μg/mL), 37˚C | This study |
| *Staphylococcus aureus* | NewHG *ahpC* (SJF5184) | *ahpC*::tet | TSB (Tetracycline 5 μg/mL), 37˚C | This study |
| *Micrococcus luteus* | SJF4393 | ATCC 4698 (Rif$^R$) | TSB (Rifampicin 0.1 μg/mL), 30˚C | [12] |
| *Enterococcus faecalis* | OG1RF | Plasmid-free wild-type strain | BHI (France), TSB (Sheffield), 37˚C | [52] |
| *Streptococcus pyogenes* | H584 | M1T1 invasive puerperal sepsis blood isolate | Columbia horse blood agar, 37˚C | [53] |
| *Streptococcus pneumoniae* | d39Δcps | Un-encapsulated strain | Todd-Hewitt supplemented with 0.5% w/v yeast extract | Lab stock |
| *Pseudomonas aeruginosa* | PA01 | Wild-type | Luria-Bertani (LB), 37˚C | Lab stock |
| *Cryptococcus neoformans* | H99 | Lab reference strain derived from Heitman lab, H99 #1 | Grown in YPD, 28˚C, then heat-killed. | [54] |
| *Escherichia coli* | W3110 (SJF4060) | Wild-type | LB | Lab stock |
| *Roseomonas mucosa* | HS or HM | Human isolate | R2A | Ian Myles (National Institute of Health) |
| *Saccharomyces cerevisiae* | 842 (SJF 66) | Wild-type | Grown in YPD at 30˚C | Lab stock |

spectrometry confirmed), 50 μg/dose CpG-B DNA (Hycult Biotech), 1% (w/v) Alhydrogel an aluminium based adjuvant (Invivogen). Vaccines were always administered subcutaneously, as above.

For the production of ClfA, XL1 blue *E. coli* were used to produce recombinant 6xHis tagged ClfA (residues 40–559). ClfA was purified from cell lysates via nickel affinity chromatography, size exclusion chromatography and confirmed as endotoxin-free. Finally, mass spectrometry was used to verify ClfA (residues 40–559) identity.

### *In vitro* ROS challenge assays

To generate ROS-killed *M. luteus*, bacteria were treated with ROS (sodium hypochlorite, $H_2O_2$ or peroxynitrite) until colonies would no longer form on agar. An aliquot of ROS-killed suspension was then dried and weighed to determine concentration, before freezing to -20˚C in PBS.

*S. aureus* and *M. luteus* from overnight broth cultures were adjusted to an optical density at 600 nm ($OD_{600}$) of 0.05 and 0.1, respectively, in 50 mL TSB. Bacteria were grown at 37˚C (*S. aureus*) or 30˚C (*M. luteus*) for 2 h, shaking. Bacteria were harvested and resuspended in PBS to $OD_{600}$ 1.0 for *S. aureus*, or 11.3 for *M. luteus*. *S. aureus* and *M. luteus* were diluted to $2x10^5$ CFU/mL and $2x10^7$ CFU/mL, respectively, in PBS. HK and ROS-killed *M. luteus* were diluted to equivalent concentrations to live *M. luteus*, as determined by dry weight (approximately 3.2

mg/mL). 50 μL *S. aureus* was incubated in the presence of 50 μL live, HK or ROS-killed *M. luteus*, or *M. luteus* PGN (25 mg/ml), with ROS and PBS to a total volume of 200 μL. *S. aureus* was incubated alone, with or without ROS for positive and negative controls. An aliquot of live bacteria present in individual tubes was taken prior to addition of ROS to determine CFU/mL at the start of the assay. $H_2O_2$ (VWR) was used at a final concentration of 0.0077% v/v, peroxynitrite (Sigma-Aldrich) at a final concentration of 2.25 mM, methyl viologen (Sigma-Aldrich) at a final concentration of 1.8 M, and sodium hypochlorite (Fisher) at a final concentration of 0.00005% v/v. Tubes were incubated at 37˚C, shaking, for 1 h before determination of CFU/mL by serial dilutions. Experimental data was combined from 3 replicates carried out on separate days.

## Cell culture

Experiments were conducted with RAW264.7 cells (ATCC TIB-71), a leukemic murine macrophage cell line, or primary MDMs derived from human blood. RAW264.7 cells were cultivated in Dulbecco's modified Eagle's medium (DMEM) supplemented with 2 mM L-Glutamine, 100 Units/mL streptomycin, 0.1 mg/mL penicillin, and 10% v/v Foetal bovine serum (FBS) (all media components sourced from Sigma-Aldrich). RAW264.7 cells were passaged into fresh media upon reaching 70–80% confluence. All experiments carried out between passages 5 and 20.

MDMs were isolated as described previously [56]. Briefly, peripheral blood mononuclear cells were isolated by Ficoll Plaque (GE Healthcare) density centrifugation, seeded in 24 well plates at $2x10^6$ cells/well in RPMI-1640 media (Lonza) supplemented with 2 mM L-Glutamine, 10% v/v newborn calf serum (Gibco) and incubated at 37˚C, 5% $CO_2$. Non-adherent cells were removed after 24 h, and adherent cells were fed with fresh RPMI-1640 supplemented with 2 mM L-Glutamine and 10% v/v low endotoxin heat-inactivated foetal bovine serum (Biosera). MDMs were used for experiments at 14 days post-isolation. Media was replaced every 2–3 days for all cells used.

## Cell infection

Cell infection assays were carried out as similar to those described previously [12], with modification. RAW264.7 or MDM cells were seeded into 24 well plates (Corning) or white 96 well micro-clear plates (Greiner) and grown to 80% confluence. DMEM supplemented with 2 mM L-glutamine was used in experimental assays for RAW264.7 cells. Cell monolayers were washed with tissue culture PBS (Fisher) three times before infection to remove residual antibiotic.

*S. aureus* was thawed from a frozen aliquot, as for mouse experiments. Unless otherwise stated, *S. aureus* was added to cells at a multiplicity of infection (MOI) 5, and HK *M. luteus* at MOI 50.

## *S. aureus* macrophage survival and mass formation assay

For experiments concerning *S. aureus* mass formation, GFP expressing *S. aureus* was added to cells with or without HK *M. luteus* in 96 well plates and incubated at 37˚C, 5% $CO_2$ for 2.5 h. Media was removed and fresh infection media supplemented with 20 μg/mL lysostaphin (Biosynexus), alongside 100 μg/ml gentamycin (Fisher) for MDMs, was added and incubated for 0.5–1 h at 37˚C, 5% $CO_2$ to kill extracellular bacteria. Monolayers were washed with PBS three times, fresh media was added and incubated overnight at 37˚C, 5% $CO_2$. Wells were imaged at 24 h post-infection using ImageXpress Micro (Molecular Devices), using a 2x objective lens and FITC filter. Masses were analysed using MetaXpress high-content image acquisition and

analysis software (Molecular Devices) for average number of masses larger than 20–40 μm per well. Experiments into the effects of chemicals on *S. aureus* mass formation were carried in the presence of 2 μM diphenyleneiodonium chloride (DPI), 500 μM apocynin or 1 μM Mito-TEMPO, all sourced from Sigma-Aldrich, with media supplemented with solvent used for controls.

In a modification of this experiment, the ratio of *S. aureus* to HK *M. luteus* was altered. RAW264.7 cells were infected with GFP *S. aureus* (MOI 5) in the presence or absence of HK *M. luteus* at a ratio to *S. aureus* of 10 (MOI 50), 5 (MOI 25), 2.5 (MOI 12.5), 0.5 (MOI 2.5), or 0.05 (MOI 0.25). Alternatively, RAW264.7 cells were infected with GFP *S. aureus* at MOI 5, 2.5, 1.25, 0.25, 0.025 in the presence or absence of HK *M. luteus* (MOI 50).

## Hydrop dye

For experiments using Hydrop fluorescent dye (Goryo chemical), mCherry *S. aureus* was stained with Alexa Fluor 555 NHS Ester (Fisher) [51] *S. aureus* were stored on ice until addition to RAW264.7 cells in a 24 well plate. *S. aureus* (MOI 5) and/or HK *M. luteus* (MOI 50) were added to cells in the presence of 1 μM Hydrop dye. 200 nM PMA with Hydrop dye was added to cells for positive control. Wells were imaged at 30 min post-infection on a Nikon Eclipse Ti microscope was used to image cells in a climate controlled set-up (37˚C, Atmosphere: 5% $CO_2$ / 95% air) with a x20 Lambda Apo NA 0.75 phase contrast objective for brightfield or with GFP or mCherry filters, images were captured with a Andor Neo-5.5-CL3 camera. Analysis was carried out using NIS elements (Nikon) and Fiji (ImageJ). The threshold for GFP images was adjusted within Fiji to exclude background fluorescence. The same threshold was set for all images from an experiment, and this was used to measure the fluorescence levels.

## CellROX

For experiments using CellROX deep red reagent (Fisher), *S. aureus* SJF4620 was first stained with 20 μM CellROX and incubated at 37˚C for 30 min, shaking. Labelled *S. aureus* (MOI 50) were added to RAW264.7 cells in the presence or absence of HK *M. luteus*, with unlabelled GFP *S. aureus* or blank media used for controls. Cells were incubated at 37˚C, 5% $CO_2$ for 30 min before fixation with 2% w/v paraformaldehyde (Sigma) for 1 h. Following fixation, samples were washed thoroughly with PBS and stained with 300 nM DAPI (Fisher) before further PBS washes. Samples were imaged on ImageXpress Micro (Molecular Devices) using a 20x objective lens, DAPI, FITC and Cy5 filters. Images were analysed using MetaXpress high-content image acquisition and analysis software. A custom module editor was used to identify bacteria in the GFP and Cy5 filters and count the frequency of signal overlap (referred to as 'oxidised objects').

## *S. aureus* infection of macrophages timelapse

For timelapse experiments, GFP *S. aureus* in the presence or absence of HK *M. luteus* was added to RAW264.7 or MDM cells in a 24 well plate, and incubated on ice for 60 min, then at 37˚C, 5% $CO_2$ for 90 min. Antibiotic-containing media was then added to cells: 20 μg/mL lysostaphin for RAW264.7 cells, 20 μg/mL lysostaphin and 100 μg/mL gentamycin for MDMs. This was incubated at 37˚C, 5% $CO_2$ for 30 min. Wells were washed with PBS three times and replaced with fresh media. MDMs were imaged every 10 min for 18 h, while RAW264.7 cells were imaged every 20 min for 18 h. Imaging was carried out on a Nikon Eclipse Ti microscope was used to image cells in a climate-controlled set-up (37˚C, Atmosphere: 5% $CO_2$ / 95% air) with a x20 Lambda Apo NA 0.75 phase contrast objective for brightfield or with a GFP filter,

images were captured with a Andor Neo-5.5-CL3 camera. Analysis was carried out using NIS elements (Nikon) and Fiji (ImageJ).

## Statistical analysis

Statistical analysis was carried out in Prism 8.4.3 (GraphPad), with $P < 0.05$ considered significant. Mouse experiments were analysed using two-tailed Mann-Whitney U tests or Kruskal-Wallis one-way analysis of variance (ANOVA) tests with Dunn's post-test, depending on the number of groups compared. Zebrafish experiments were analysed using unpaired t-tests. *In vitro* ROS challenge assays were analysed using one-way ANOVAs with Tukey post-test. Cell infection assays were analysed by Mann-Whitney U tests or Kruskal-Wallis ANOVA tests with Dunn's post-test. All measurements were taken from distinct samples.

## Supporting information

**S1 Fig. Breadth of the augmentation phenomenon. A** Co-injection of low dose *S. aureus* (SA $1 \times 10^6$ CFU) with heat-killed *M. luteus* (HK ML equivalent of $1 \times 10^8$ CFU) into mice: weight loss (n = 5 per group), $^*p < 0.05$ **B** Co-injection of low dose *S. aureus* (SA $1 \times 10^6$ CFU) with *E. coli* (EC $5 \times 10^6$ CFU) into mice: weight loss (n = 8–10 per group, *S. aureus*, circles; *E. coli* triangles), $^{**}p < 0.003$; $^{***}p < 0.0005$ **C** Co-injection of low dose *S. aureus* (SA $1 \times 10^6$ CFU) with *R. mucosa* (RM $2 \times 10^8$ CFU) into mice: weight loss (n = 5 per group) **D** Co-injection of low dose *S. aureus* (SA $1 \times 10^6$ CFU) with heat-killed *C. neoformans* (CN 750 μg) into mice: weight loss (n = 9–10 per group) **E-G** Co-injection of low dose *S. aureus* (SA $1 \times 10^6$ CFU) with *S. cerevisiae* (SC $1 \times 10^8$ CFU) into mice, (n = 10 per group, *S. aureus*, circles; *S. cerevisiae* triangles): **E** weight loss, $^*p < 0.05$; $^{****}p < 0.0001$, **F** liver CFU, $^{****}p < 0.0001$, **G** kidney CFU, $^{***}p < 0.0005$ **H-L** Co-injection of low dose *E. faecalis* (EF $5 \times 10^7$ CFU) with *M. luteus* (ML $2 \times 10^8$ CFU) into mice: **H** weight loss (n = 10 per group), **I** kidney CFU, **J** lung CFU, $^*p \leq 0.05$, **K** heart CFU, **L** spleen **M-N** Co-injection of low dose *E. faecalis* (EF $5 \times 10^7$ CFU) with *M. luteus* PGN (ML PGN 500 μg) into mice (n = 10 per group): **M** weight loss, $^*p < 0.05$, **N** liver CFU, **O-R** Co-injection of high dose *E. faecalis* (EF $1 \times 10^8$ CFU) with *M. luteus* PGN (ML PGN 500 μg) into mice, CFUs taken at 48 hpi (n = 8–9 per group): **O** liver CFU, $^{****}p < 0.0001$, **P** kidney CFU, $^{**}p < 0.0003$, **Q** spleen CFU, $^{***}p < 0.0005$, **R** heart CFU, $^*p < 0.05$. Colours used indicate the level of significance as indicated in Fig 1E. For all panels, the median value is shown. For panels B and E, a Kruskal-Wallis test with Dunn's post-test was used, for all remaining panels a two-tailed Mann-Whitney test was used. CFUs were enumerated at 3 days post-infection, unless otherwise stated.
(TIF)

**S2 Fig. Lack of augmentation of alternative bacterial pathogens. A-C** Co-injection of low dose *S. pneumoniae* (SPN $1 \times 10^6$ CFU) with *M. luteus* PGN (ML PGN 500 μg) into mice (n = 10 per group), CFUs enumerated at 3 days post-infection: **A** weight loss, $^*p < 0.05$, **B** liver CFU, **C** kidney CFU, $^{**}p < 0.003$ **D-F** Co-injection of low dose *P. aeruginosa* (PA $1 \times 10^5$ CFU) with *M. luteus* PGN (ML PGN 500 μg) into mice (n = 10 per group), CFUs enumerated at 3 days post-infection: **D** weight loss, **E** liver CFU, **F** kidney CFU **G-H** Intramuscular co-injection of *S. pyogenes* (SPY $1 \times 10^8$ CFU) with *M. luteus* PGN (ML PGN 500 μg) into mice (n = 8 per group): **G** leg hindlimb CFU at 24 hpi, **H** weight loss at 24 hpi. Colours used indicate the level of significance as indicated in Fig 1E. For all panels, a two-tailed Mann-Whitney test was used and the median value is shown.
(TIF)

**S3 Fig. Augmenting material must be present at the same time and location as *S. aureus* to enhance infection. A-B** Mice were intravenously injected with 500 μg *M. luteus* PGN 24 hours, 6 hours or 1 hour before infection with $1 \times 10^6$ *S. aureus*, at the same time as *S. aureus*, or with *S. aureus* alone (n = 7 per group, median value shown, Kruskal-Wallis tests with Dunn's post-test) **A** weight loss **B** kidney CFU **C-D** Mice were intravenously injected with 500 μg *M. luteus* PGN 48 hours, 24 hours or 6 hours after infection with $1 \times 10^6$ *S. aureus*, at the same time as *S. aureus*, or with *S. aureus* alone (n = 5 per group, median value shown, Kruskal-Wallis tests with Dunn's post-test) **C** weight loss, $^*p < 0.05$; $^{**}p < 0.008$ **D** kidney CFU **E** quantitation of zebrafish macrophages from Fig 2I, showing the percentage containing *S. aureus* only (green), PGN only (blue) or *S. aureus* co-localising with PGN (yellow) (n = 47 macrophages from 21 larvae).
(TIF)

**S4 Fig. Augmenting material protects *S. aureus* from ROS *in vitro*. A** Following methyl viologen exposure, percentage survival of *S. aureus* alone ($5 \times 10^4$ CFU/mL), or with *M. luteus* PGN (1.25 mg) (n = 3, error bars show mean +/- SD, one-way ANOVA test with Tukey's post hoc test), $^*p < 0.05$; $^{**}p < 0.01$ **B-D** Following ROS exposure, percentage survival of *S. aureus* alone ($5 \times 10^4$ CFU/mL), or with live *M. luteus* ($5 \times 10^6$ CFU/mL), heat-killed *M. luteus* (equivalent of $5 \times 10^6$ CFU/mL) or ROS killed *M. luteus* (equivalent of $5 \times 10^6$ CFU/mL), (n = 3, error bars show mean +/- SD, one-way ANOVA tests with Tukey's post hoc test) **B** methyl viologen, $^{****}p < 0.0001$ **C** hydrogen peroxide, $^{**}p < 0.007$ **D** peroxynitrite, $^{**}p < 0.003$; $^{****}p < 0.0001$ **E** Following sodium hypochlorite exposure, percentage survival of *S. aureus* alone ($5 \times 10^4$ CFU/mL), or with live *M. luteus* ($5 \times 10^6$ CFU/mL), (n = 3, error bars show mean +/- SD, one-way ANOVA test with Tukey's post hoc test), $^*p < 0.05$.
(TIF)

**S5 Fig. ROS susceptible *S. aureus* survives ROS exposure due to protection by augmenting material. A-B** Following hydrogen peroxide exposure, percentage survival of ROS susceptible *S. aureus* mutants alone ($5 \times 10^4$ CFU/mL), or with live *M. luteus* ($5 \times 10^6$ CFU/mL), heat-killed *M. luteus* (equivalent of $5 \times 10^6$ CFU/mL) or $H_2O_2$ killed *M. luteus* (equivalent of $5 \times 10^6$ CFU/mL), (n = 3, error bars show mean +/- SD, one-way ANOVA tests with Tukey's post hoc test): **A** *katA*, $^*p < 0.05$ **B** *ahpC*, $^*p < 0.05$; $^{***}p < 0.0007$ **C** Following sodium hypochlorite exposure, percentage survival of *S. aureus katA ahpC* alone ($5 \times 10^4$ CFU/mL), or with live *M. luteus* ($5 \times 10^6$ CFU/mL), heat-killed *M. luteus* (equivalent of $5 \times 10^6$ CFU/mL) or ROS killed *M. luteus* (equivalent of $5 \times 10^6$ CFU/mL), (n = 3, error bars show mean +/- SD, one-way ANOVA test with Tukey's post hoc test), $^{****}p < 0.0001$ **D-F** Injection of high dose ($1 \times 10^7$) of *S. aureus* control, *katA*, *katA ahpC* into mice (n = 10 per group, median value shown, Kruskal-Wallis tests with Dunn's post-test): **D** weight loss, $^{**}p < 0.009$; $^{***}p < 0.0003$ **E** liver CFU, $^*p < 0.05$ **F** kidney CFUs, $^*p < 0.05$; $^{**}p < 0.01$ **G-I** Injection of high dose ($1 \times 10^7$) of *S. aureus* control and *ahpC* into mice (n = 10 per group, median value shown, two-tailed Mann-Whitney tests): **G** weight loss, $^*p < 0.05$ **H** liver CFU, $^{***}p < 0.0002$ **I** kidney CFU **J-K** Co-injection of low dose ($1 \times 10^6$ CFU) *S. aureus katA ahpC* with heat-killed *M. luteus* (equivalent of $1 \times 10^8$ CFU) into mice: liver CFU (n = 5 per group, median value shown, two-tailed Mann-Whitney tests): **J** weight loss **K** kidney CFU **L** Co-injection of low dose *S. aureus* (SA $2 \times 10^5$ CFU) with *M. luteus* PGN (ML PGN 500 μg) into wild-type (WT) control mice or MPO$^{-/-}$ mice: weight loss (n = 9 per group, median value shown, individual two-tailed Mann-Whitney tests), $^*p < 0.05$; $^{**}p < 0.004$.
(TIF)

**S6 Fig. Augmenting material protects *S. aureus* from ROS in murine and human macrophages. A** Representative images of human MDMs infected with GFP *S. aureus* (MOI 5) in the

presence or absence of heat-killed *M. luteus* (MOI 50), showing number of GFP *S. aureus* masses observed (n = 3, individual images show a single field of view, 4 per well), scale 100 μm **B** *S. aureus*-GFP growth curve in the presence of 0, 2, 5 or 10 μM DPI **C** *S. aureus*-GFP growth curve in the presence of 500 μM apocynin or solvent control **D** Representative images of RAW264.7 cells in the presence of Hydrop (1 μM) (green) incubated alone (media), treated with PMA (200 nM), or infected with *S. aureus* (MOI 5), heat-killed *M. luteus* (MOI 50), or both *S. aureus* and heat-killed *M. luteus*, scale 20 μm **E** Representative images of RAW264.7 cells incubated alone (media), infected with *S. aureus* (MOI 50), CellROX-stained *S. aureus* (MOI 50) or CellROX-stained *S. aureus* (MOI 50) and heat-killed *M. luteus* (MOI 50). RAW264.7 nuclei were stained with DAPI (blue), *S. aureus* was labelled by GFP expression (green) and, upon oxidation, by CellROX (red). Arrows indicate co-localisation of GFP *S. aureus* with CellROX signal, implying oxidation of bacteria, scale 50 μm. White box indicates area of increased magnification, scale 10 μm.
(TIF)

**S7 Fig. Vaccination reduces augmented infection. A-B** Mice were vaccinated subcutaneously on day 0, 14 and 21 with vaccine (1 μg ClfA, 50 μg CpG and 1% w/v Alhydrogel, triangles) or PBS control (circles). Day 28 post-vaccination mice were intravenously injected with low dose *S. aureus* ($1x10^6$ CFU), high dose *S. aureus* ($1x10^7$ CFU), or both low dose *S. aureus* and 250 μg *M. luteus* PGN (n = 10 per group, median value shown, two-tailed Mann-Whitney tests) **A** weight loss, $^*p<0.05$ **B** kidney CFU, $^*p<0.05$ **C-F** Mice were vaccinated subcutaneously on day 0, 14 and 21 with vaccine (1 μg ClfA, 50 μg CpG and 1% w/v Alhydrogel, triangles) or PBS control (circles). Day 28 post-vaccination mice were intravenously injected with high dose *S. aureus* ($5x10^6$ CFU), or low dose *S. aureus* ($5x10^5$ CFU) and 250 μg *M. luteus* PGN (n = 10 per group, median value shown, two-tailed Mann-Whitney tests) **C** diagram of experimental protocol **D** liver CFU, $^{****}p<0.0001$ **E** weight loss, $^*p<0.05$; $^{**}p<0.006$ **F** kidney CFU.
(TIF)

**S8 Fig. Diagram of augmentation phenomenon mechanism.** Diagram highlighting how augmenting material protects *S. aureus* from ROS in the phagosome of macrophages. The left-hand side demonstrates non-augmented *S. aureus* infection, resulting in ROS mediated bacterial killing. The right-hand side shows an augmented *S. aureus* infection, highlighting how the presence of augmenting material in the same phagosome as *S. aureus* results in reduced *S. aureus* killing due to inactivating ROS produced in the phagosome. Survival *S. aureus* is then able to proliferate and escape the macrophage.
(TIF)

# Acknowledgments

We thank Dr Lynda Partridge for the use of tissue culture and cell infection facilities. We thank Joseph Murphy for assistance with *in vitro* assays. We thank Stephen Brown and Lucie N'Koy at the Sheffield RNAi Screening Facility, Biomedical Sciences, University of Sheffield for providing training and equipment used in this study. We thank The Bateson Centre Aquarium Facility and the Biological Services Unit staff at The University of Sheffield for technical support during the *in vivo* experiments. Imaging was performed in the Wolfson Light Microscopy Facility using a Perkin Elmer spinning disk imaging system. ADB, CA, PS thank M.A. Nahori for technical advice and thank staff members from the conventional animal facilities (IERP, INRA, 78352 Jouy-en-Josas, France). For the purpose of open access, the authors have applied a CC BY public copyright licence to any Author Accepted Manuscript version arising from this submission.

## Author Contributions

**Conceptualization:** Josie F. Gibson, Grace R. Pidwill, Stephen A. Renshaw, Simon J. Foster.

**Data curation:** Josie F. Gibson, Grace R. Pidwill, Oliver T. Carnell, Bas G. J. Surewaard, Daria Shamarina.

**Formal analysis:** Josie F. Gibson, Grace R. Pidwill, Oliver T. Carnell, Bas G. J. Surewaard, Daria Shamarina, Joshua A. F. Sutton, Charlotte Jeffery, Aurélie Derré-Bobillot, Cristel Archambaud, Matthew K. Siggins, Eric J. G. Pollitt, Simon A. Johnston.

**Funding acquisition:** Shiranee Sriskandan, Stephen A. Renshaw, Simon J. Foster.

**Investigation:** Josie F. Gibson, Grace R. Pidwill, Oliver T. Carnell, Bas G. J. Surewaard, Daria Shamarina, Joshua A. F. Sutton, Charlotte Jeffery, Aurélie Derré-Bobillot, Cristel Archambaud, Matthew K. Siggins, Eric J. G. Pollitt, Simon A. Johnston.

**Methodology:** Josie F. Gibson, Grace R. Pidwill, Oliver T. Carnell, Bas G. J. Surewaard, Daria Shamarina, Joshua A. F. Sutton, Charlotte Jeffery, Aurélie Derré-Bobillot, Cristel Archambaud, Matthew K. Siggins, Eric J. G. Pollitt, Simon A. Johnston.

**Project administration:** Stephen A. Renshaw, Simon J. Foster.

**Supervision:** Pascale Serror, Shiranee Sriskandan, Stephen A. Renshaw, Simon J. Foster.

**Writing – original draft:** Josie F. Gibson, Grace R. Pidwill.

**Writing – review & editing:** Josie F. Gibson, Grace R. Pidwill, Bas G. J. Surewaard, Simon A. Johnston, Pascale Serror, Shiranee Sriskandan, Stephen A. Renshaw, Simon J. Foster.

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
