## [Decision Letter · Decision Letter 0]

7 Jul 2021

Dear Prof. Foster,

Thank you very much for submitting your manuscript "Commensal bacteria augment Staphylococcus aureus infection by inactivation of phagocyte-derived reactive oxygen species" for consideration at PLOS Pathogens. As with all papers reviewed by the journal, your manuscript was reviewed by members of the editorial board and by several independent reviewers. The reviewers appreciated the attention to an important topic. Based on the reviews, we are likely to accept this manuscript for publication, providing that you modify the manuscript according to the review recommendations.

As indicated by the comment of the reviewers, the data in the manuscript are of substantial interest and well presented. Each of the reviewers has a number of concerns and/or clarifications which must be addressed. In particular, reviewer 1 asks, rightly so, whether just increasing the dose of S. aureus itself would similarly function as a sponge for ROS - and why this is not the case. Similarly, rev 3 makes important points regarding the actual mechanism of ROS "neutralization" and where this occurs. Overall, the concept that macrophage generation of ROS is critical in S. aureus clearance is important and this demonstration of heterogeneous augmentation is a nice contribution.

Sincerely,

Alice Prince

Associate Editor

PLOS Pathogens

Michael Otto

Section Editor

PLOS Pathogens

Kasturi Haldar

Editor-in-Chief

PLOS Pathogens

orcid.org/0000-0001-5065-158X

Michael Malim

Editor-in-Chief

PLOS Pathogens

orcid.org/0000-0002-7699-2064

As indicated by the comment of the reviewers, the data in the manuscript are of substantial interest and well presented. Each of the reviewers has a number of concerns and/or clarifications which must be addressed. In particular, reviewer 1 asks, rightly so, whether just increasing the dose of S. aureus itself would similarly function as a sponge for ROS - and why this is not the case. Similarly, rev 3 makes important points regarding the actual mechanism of ROS "neutralization" and where this occurs. Overall, the concept that macrophage generation of ROS is critical in S. aureus clearance is important and this demonstration of heterogeneous augmentation is a nice contribution.

Reviewer Comments (if any, and for reference):

Reviewer's Responses to Questions

**Part I - Summary**

Reviewer #1: In this study the authors extend upon their earlier work that documented a process referred to as augmentation whereby skin commensals along with peptidoglycan can enhance the pathogenesis of S. aureus. The authors provide evidence that this process occurs through ameliorating the ROS/killing mechanism of macrophages. This can occur with live and heat killed preparations as well as peptidoglycan. This aspect of the study is well executed. The other elements of the work pertaining to what is required for augmentation and how this inhibits ROS activity is less developed. Other than a multitude of species tested, it is not known if additional S. aureus or HK S. aureus would cause this augmentation phenomenon or the scope of different bacterial components in protection S. aureus from ROS damage. If indeed the augmenting material is just acting as a decoy to ROS, this is a straight forward mechanism, in the sense that more cells will require more time to be killed. Both of these points need some further investigation, along with the vaccination data at the end which doesn’t add to the mechanistic data presented regarding the ROS aspect of the study.

Reviewer #2: Could it be that S. cervisiae did not augment S. aureus counts in the liver and S. pneumoniae and P. aeruginosa those of E. faecalis, because these three species may target other organs or body districts. Could authors provide complementary counts of other organs and blood for the supplementary materials?

The hypothesis of augmentation envisages that in each phagosome of each phagocyte the augmenting material co-localises with S. aureus or S. aureus katA aphC. For example in the case of the liver samples (time of sampling in fig 1 not defined) which might be at 72h it is not clear if authors hypothesise that liver macrophages did take up at time 1h post-infection one S. aureus cell and one M. luteus and that both remained together within the same macrophage for 72h or if authors hypothesise that at each cycle of liver macrophage infection, which might occur each 6h, each successive event is determined by a colocalization of S. aureus and M. luteus. There might also be the hypothesis that the colocalization is necessary only for the first moment and that from then on S. aureus is able to start the successive phases of disease (not defined here) also, without needing the help of the mcirocossus. The last paragraph of the discussion nicely addresses this, but authors should provide references for the discussion of their hypotheses.

Please define time of sampling (i.e. liver) of all animal experiments (i.e. Fig 1 and 3).

Figure 2i authors should give a quantitative or at least semi-quantitative analysis which shows if the phagosome shown in 2i was the only one showing effectively colocalization or if this was the case in many phagocytes.

Reviewer #3: Gibson et al. investigate the ability of commensal bacteria or material from commensal bacteria to promote S. aureus survival and pathogenicity. The authors show that this phenomenon, named augmentation, is caused by the ability of commensal bacteria and associated materials to blunt the antimicrobial effect of reactive oxygen species. In general, the findings are interesting and an advance for the field. I have a few comments for the authors to consider (below).

**Part II – Major Issues: Key Experiments Required for Acceptance**

Reviewer #1: It was shown with different pathogens that they could all augment the S. aureus infection. Beyond what the authors have shown before with PGN and other skin commensals, this does not add further mechanistic data. Can other PAMPS also augment the response? Alternatively, is there a generalized host signaling pathway that is stimulated by all the different species tested that influences the outcome. Does adding back additional S. aureus PGN also augment?

Is the augmentation material just acting as a sponge/decoy to the immune system and reactive oxygen species? The data presented in figure 3 seems to support this. If this is the case, wouldn’t just adding heat-killed S. aureus or even more S. aureus cause the same effect? If there is a finite ability of the immune system and/or ROS to deal with bacterial cells adding more would act to quench the activity.

It is known, and acknowledged that murine models of S. aureus while showing promise with vaccines have yet to translate to human studies. While it was interesting to observe that the vaccine used only appeared to cause an effect with the augmented sample, it would be expected based on prior studies that the formulation used would also protect against the standard challenge.

In figure 4b the WT control is missing. The levels of S. aureus survival of the katAahpC mutant look similar to what was seen with WT in other experiments, without this control, the mutant’s inherent attenuation cannot be confirmed.

Figure 3, much of the data is only an n of 3. It is not clear if this data is the accumulation from multiple experiments, or if the experiment needs reproducing.

Reviewer #2: Availability of microscopy of early liver samples showing colocalization would strengthen the data.

Reviewer #3: (No Response)

**Part III – Minor Issues: Editorial and Data Presentation Modifications**

Reviewer #1: In figure 2, the color that is used to track PGN is not noted. It is also hard to see if the PGN added is indeed co-localised with S. aureus.

Several pertinent comparisons in the data were lack statistical testing, such as Figure 3b is S. aureus alone different to S. aureus with PGN in the present of H2O2? Likewise, 3f, comparing the different groups with H2O2 treatment.

In figure 6, the level of infection with both the low and high doses of S. aureus without vaccination were less than what was observed in earlier experiments.

The authors may wish to include in their discussion why they see protection/augmentation with heat-killed preparations under some conditions but not others.

Reviewer #2: none

Reviewer #3: 1. While it is accurate to state the augmentation phenotype is not observed in transgenic mice lacking a functional NOX2 (as shown in Fig. 4, Boldock et al., Nat Microbiol, 2018), the text related to the requirement for NOX2 in augmentation needs clarification in the current manuscript. Simply put, the shielding phenotype conferred by commensal microbes would not be needed to promote optimal growth in mice lacking NOX2. The growth of SA alone is augmented already because phagocytes in Cybb-/- mice lack the ability to produce ROS, which contribute to bacterial killing. This is the likely reason for a lack of difference between SA alone and SA + PGN in Cybb-/- mice (Fig 4b) in the Boldock et al. Nat Microbiol paper. This point is underscored further by the much greater RFL per mm2 in Cybb-/- mice compared to that in the wild-type mice. These findings support the authors conclusions in the current manuscript, but not for the reasons implied by the text (lines 156 and 187). That is, ROS do not contribute to the shielding mechanism in vitro, but they are required to illustrate a difference between shielded and unshielded phenotypes. There could also be a difference in mechanism between in vitro assays and in vivo models, although this is beyond the scope of the current studies. Please revise the text to clarify the conclusion related to the role of NOX2, which was not optimally stated.

2. The assays that test the effects of reactive oxygen species in vitro can be influenced considerably by the assay buffer or media. For example, hydrogen peroxide can kill S. aureus readily in PBS, as shown in Fig. 3. However, the results will be different in an assay buffer that permits sustained S. aureus metabolism and/or replication, even if minimal (e.g., DMEM with glucose). Also, hydrogen peroxide (at millimolar concentrations) can have little or no impact on S. aureus survival in a rich media such as TSB. This is likely due to the interaction of ROS with components of media (a mechanism perhaps similar to the “shielding” reported here), and the ability of S. aureus to rapidly produce molecules needed for defense against ROS. This does not occur in PBS. I understand that these scenarios are different. The PBS assay addresses whether ROS have the potential to kill S. aureus, whereas assays performed in tissue culture or bacterial growth media address whether ROS kill S. aureus under conditions that permit sustained S. aureus metabolism and growth. Some additional text might be help clarify this potential issue with in vitro assays.

PLOS authors have the option to publish the peer review history of their article (what does this mean?). If published, this will include your full peer review and any attached files.

Reviewer #1: No

Reviewer #2: **Yes: **Marco Rinaldo Oggioni

Reviewer #3: No

Figure Files:

Data Requirements:

Reproducibility:

References:

---

## [Editor Report · Decision Letter 1]

9 Aug 2021

Dear Prof. Foster,

We are pleased to inform you that your manuscript 'Commensal bacteria augment Staphylococcus aureus infection by inactivation of phagocyte-derived reactive oxygen species' has been provisionally accepted for publication in PLOS Pathogens.

Best regards,

Alice Prince

Associate Editor

PLOS Pathogens

Michael Otto

Section Editor

PLOS Pathogens

Kasturi Haldar

Editor-in-Chief

PLOS Pathogens

orcid.org/0000-0001-5065-158X

Michael Malim

Editor-in-Chief

PLOS Pathogens

orcid.org/0000-0002-7699-2064
---

## [Editor Report · Acceptance letter]

10 Sep 2021

Dear Prof. Foster,

We are delighted to inform you that your manuscript, "Commensal bacteria augment Staphylococcus aureus infection by inactivation of phagocyte-derived reactive oxygen species," has been formally accepted for publication in PLOS Pathogens.

Best regards,

Kasturi Haldar

Editor-in-Chief

PLOS Pathogens

orcid.org/0000-0001-5065-158X

Michael Malim

Editor-in-Chief

PLOS Pathogens

orcid.org/0000-0002-7699-2064